# Liposomes as Imaging Agents of Inflammation and Oxidative Stress in Bone Implants

**DOI:** 10.3390/cimb47050295

**Published:** 2025-04-22

**Authors:** Delia Danila, Patricia S. Pardo, R. Devesh Kumar Misra, Aladin M. Boriek

**Affiliations:** 1Graduate School of Biomedical Sciences, Baylor College of Medicine, Houston, TX 77030, USA; delia.danila@bcm.edu; 2Department of Medicine, Baylor College of Medicine, Houston, TX 77030, USA; pardopatriciasusa@gmail.com; 3Department of Metallurgical, Materials, and Biomedical Engineering, University of Texas at El Paso, El Paso, TX 79968, USA; dmisra2@utep.edu

**Keywords:** liposomes, non-invasive imaging agents, bone-implant complications, inflammation, oxidative stress, bioimaging

## Abstract

Liposomes are tiny, spherical vesicles made from cholesterol and natural phospholipids that are promising imaging agents for detecting medical complications. They can carry fluorescent markers or other imaging agents, making them effective for medical imaging. Furthermore, liposomes can target specific cells involved in inflammation, such as macrophages, and accumulate at inflammation sites when injected. Additionally, liposomes can be designed to respond to oxidative stress, which is often associated with bone implant complications. By detecting areas of stress, liposomes provide valuable information about implant health. However, challenges such as rapid clearance from the body, precise targeting, immune reactions, and high production costs must be addressed. Research is ongoing to improve the design and functionality of liposomes. They can potentially monitor bone implants as non-invasive imaging agents, enabling early detection of complications and timely interventions. This approach can enhance patient outcomes and extend the longevity of implants, making it a promising strategy for better patient care and implant success.

## 1. General Introduction to Biodegradable Bone Implants

Metal implants are commonly used for bone healing, with materials such as stainless steel, pure titanium, titanium-based alloys, and biodegradable options like magnesium alloys being the most prevalent. Biodegradable metal alloys are increasingly recognized for their ability to degrade after bone healing, eliminating the need for secondary surgery to remove them [1].

Magnesium alloys are mainly studied for their mechanical properties, which closely resemble those of bone in terms of low elasticity compared to conventional implants. Findings from in vivo clinical trials have shown that magnesium degradation promotes bone growth by stimulating increased osteoblastic activity [2]. However, it is essential to acknowledge that magnesium alloys can corrode more quickly than the repair or healing process. Ideally, the implant should degrade over time in tandem with the healing and remodeling process, as illustrated in Figure 1 [2].

Magnesium alloys possess a density of approximately 1.7 g/cm^3^, making them significantly lighter than aluminum and steel [3]. This low density, combined with their strength, gives them an outstanding strength-to-weight ratio. One type of magnesium alloy with increased corrosion resistance is Mg-xGd. Adding up to 10 wt% Gadolinium improves the mechanical properties of magnesium by promoting a slower degradation rate [4,5]. The table below (Table 1) outlines the advantages and disadvantages of using magnesium alloys for orthopedic implants [2].

Pre-clinical animal studies can provide essential insight into using Mg-based implants and their subsequent use in clinical studies [6].

## 2. Imaging Techniques That Look at Bone Healing After Implantation

X-ray scattering, X-ray diffraction, and Magnetic Resonance Imaging (MRI) are the imaging techniques explored for assessing bone healing. Implantation can lead to an inflammatory response. A variety of cells are recruited around the implantation site. In the initial 1–7 days following implantation, there is an increase in immune cells. During the first 0–2 days post-injury, neutrophils migrate to the injury site, followed by the aggregation of macrophages, a key indicator of early-stage inflammation [7]. The innate immune system is vital for initiating inflammatory reactions, and this activation in response to metal implantation can be investigated [8,9].

While inflammation is a normal response to implantation and can help with the healing process, excessive inflammation can lead to fibrosis [10].

The best way to study the inflammatory response is through non-invasive in vivo imaging methods. Several non-invasive imaging methods have been used in preclinical studies to examine inflammation related to the implant. One of these non-invasive imaging modalities is optical imaging. Recent advances in the development of biocompatible near-infrared fluorochromes, as well as the development of targeted imaging agents, have made optical imaging an increasingly attractive imaging tool. Optical imaging has the potential to identify macrophages at the site of inflammation [1]. Other non-invasive imaging techniques include magnetic resonance imaging (MRI), nuclear imaging, and computed tomography (CT). Among all these imaging modalities, CT offers several advantages, such as exceptional spatial resolution and deep tissue penetration [11]. Another way to monitor these processes is by coupling optical imaging, which gives us information regarding the immune responses, with micro-CT, which provides us with information on the implants’ degradation.

Currently, there is a lack of biocompatible contrast agents designed to detect and/or visualize inflammatory processes related to bone healing [11].

## 3. Nanomaterials as Imaging Agents

Nanomaterials seem promising as contrast agents and as carriers for imaging agents or drugs. Their size can vary from 1 nm to 100 nm or higher. Their surface can be modified to target them to a specific site [12,13]. They can also reduce the toxicity of the contrast agent or drug and increase their solubility. The most well-known nanomaterials used are liposomes, polymeric nanoparticles, micelles, nanocrystal nanoparticles, and inorganic nanoparticles [14].

Key parameters for nanomaterials include size, shape, circulation duration, and surface characteristics. An essential parameter is the size of the nanomaterials, since it affects the uptake, biodistribution, and encapsulation efficiency. The optimal size for a nanoparticle depends on the cellular uptake and targeting availability. Nanoparticles that are too small (smaller than 10 nm) or too big (bigger than 200 nm) are cleared fast by the reticuloendothelial system (RES). Small particles are quickly eliminated by the kidneys, and the bigger nanoparticles accumulate in the liver and spleen. The optimal size of the nanoparticles varies depending on the application and should not exceed 200 nm. The shape also matters; while many nanoparticles are spherical, others can be rod-shaped or cylindrical. Surface modifications are essential for directing the nanoparticles to a specific target site or ensuring prolonged circulation. Liposomes are among the most effective nanoparticles to target and effectively deliver contrast agents/drugs/genes to cells [13].

## 4. Liposomes and Therapeutic Strategies

Liposomes are highly biocompatible nanoparticles that can be targeted and can carry imaging agents/drugs to inflammation sites [14]. Depending on how they are formulated, liposomes can be used as imaging agents for various imaging techniques [15,16].

Introduced by Bangham in 1961, liposomes are models of biological membranes made from lipids. They self-assemble into spherical structures or vesicles with a polar interior designed to encapsulate polar agents, a non-polar hydrophobic lipid bilayer for loading non-polar agents, and a polar surface that can be functionalized for extended circulation or targeting. Liposomes can deliver the encapsulated drug through passive targeting or by active targeting. In passive targeting, they fuse with the biological membrane and deliver their payload. In active targeting, ligands are attached to the liposome surface to specifically recognize and bind to disease markers. A schematic representation of the different types of liposomes is presented in Figure 2. One type of liposomes is conventional liposomes. They are non-functionalized liposomes that can carry a payload (Figure 2, upper left quadrant). Another type of liposomes is PEGylated liposomes. These liposomes are coated with polyethylene glycol. They are engineered to remain in circulation for longer and are commonly referred to as “stealth” liposomes (Figure 2, upper right quadrant). A third type of liposomes is ligand-targeted liposomes. These liposomes are functionalized to target specific sites by incorporating a specific ligand to which antibodies, peptides, and proteins can bind (Figure 2, lower right quadrant). The multifunctional liposomes are the liposomes that have more functions, and they can be used for diagnostic or treatment purposes (Figure 2, lower left quadrant). These liposomes carry the desired contrast agent or drug, are long-circulating, and are functionalized to bind to the disease site specifically.

Liposomes are primarily composed of phospholipids, with additional materials added for stability, targeting, and detection. Cholesterol is often included to modify the membrane permeability. There are several ways to prepare liposomes, such as lipid film hydration, sonication, freeze-drying, solvent injection, and detergent dialysis. There are advantages and disadvantages to each preparation method. The liposomes synthesized by these different methods will be either multilamellar (sonication, freeze-drying, solvent injection, detergent dialysis) or unilamellar (lipid film hydration with extrusion). The different methods involve some common steps, such as the formation of a lipid film and the hydration of the lipid film [12]. One standard method for preparing reproducible liposomes that will render unilamellar liposomes of the same size involves the hydration of a thin lipid film followed by extrusion. The lipid film is made by evaporating the chloroform from the solution of lipids dissolved in chloroform, as illustrated in Figure 3 [12]. If the contrast agent or drug is hydrophilic, it is added during the hydration step. For hydrophobic agents or drugs, they are incorporated into the lipid solution in chloroform before forming the thin lipid film.

Since 1995, various liposomal drug formulations have been commercialized and approved in medical practice, as it can be seen in Table 2 [18]. The first clinically approved liposomal formulation was Doxil for the treatment of ovarian cancer, breast cancer, and Kaposi’s sarcoma. There are other liposomal formulations approved for the treatment of cancer, such as DaunoXome, Myocet, and Marqibo. Other liposomal drug formulations, such as Abelcet, Ambisome, and Amphotec, were designed to treat fungal infections, while others were approved for pain management, such as DepoDur and Exparel. One of the most recent liposomal formulations, Inflexal, was approved for the treatment of influenza. The liposomal formulations are delivered either intravenously or intramuscularly. The active agent is encapsulated in liposomes.

**Table 2 cimb-47-00295-t002:** Clinically used liposome-based products [18].

SN	Clinical Products (Approval Year)	Administration	Active Agent	Indication	Company
1.	Doxil^®^(1995)	i.v.	Doxorubicin	Ovarian cancer, breast cancer, Kaposi’s sarcoma	Sequus Pharmaceuticals
2.	DaunoXome^®^(1996)	i.v.	Daunorubicin	AIDS-related Kaposi’s sarcoma	NeXstar Pharmaceuticals
3.	Depocyt^®^(1999)	Spinal	Cytarabine/Ara-C	Neoplastic meningitis	SkyPharma Inc.
4.	Myocet^®^(2000)	i.v.	Doxorubicin	Combination therapy with cyclophosphamide in metastatic breast cancer	Elan Pharmaceuticals
5.	Mepact^®^(2004)	i.v.	Mifamurtide	High-grade, resectable, non-metastatic osteosarcoma	Takeda Pharmaceutical Limited
6.	Marqibo^®^(2012)	i.v.	Vincristine	Acute lymphoblastic leukaemia	Talon Therapeutics, Inc.
7.	Onivyde™ (2015)	i.v.	Irinotecan	Combination therapy with fluorouracil and leucovorin in metastatic adenocarcinoma of the pancreas	Merrimack Pharmaceuticals Inc.
8.	Abelcet^®^(1995)	i.v.	Amphotericin B	Invasive severe fungal infections	Sigma-Tau Pharmaceuticals
9.	Ambisome^®^ (1997)	i.v.	Amphotericin B	Presumed fungal infections	Astellas Pharma
10.	Amphotec^®^ (1996)	i.v.	Amphotericin B	Severe fungal infections	Ben Venue Laboratories Inc.
11.	Visudyne^®^ (2000)	i.v.	Verteporphin	Choroidal neovascularisation	Novartis
12.	DepoDur™ (2004)	Epidural	Morphine sulfate	Pain management	SkyPharma Inc.
13.	Exparel^®^(2011)	i.v.	Bupivacaine	Pain management	Pacira Pharmaceuticals, Inc.
14.	Epaxal^®^(1993)	i.m.	Inactivated hepatitis A virus (strain RGSB)	Hepatitis A	Crucell, Berna Biotech
15.	Inflexal^®^ V (1997)	i.m.	Inactivated hemaglutinine of Influenza virus strains A and B	Influenza	Crucell, Berna Biotech

i.v. (intravenous); i.m. (intramuscular).

## 5. Liposomes as an Imaging Agent for Bone Implants

Liposomes can be used as molecular imaging agents in conjunction with CT scanners to monitor inflammation and oxidative stress levels around implants. The implantation of biomaterials begins with an injury, followed by blood–material interactions, provisional matrix formation, and acute inflammatory response. This initial inflammatory phase is a typical response to implanted biomaterials and is crucial for wound healing, implant integration, and ossification [15,19]. After implantation, a wound-healing reaction occurs around the prosthetic device, remodeling the surrounding tissue and promoting osteointegration. Both in vitro and in vivo data show that prosthetic bioproducts interact with innate immunity receptors on the surface of immune cells, triggering an acute inflammatory response. This response is characterized by releasing inflammatory, cytokine, chemokine, and reactive oxygen species (ROS), which aim to remodel the surrounding tissue in response to the prosthetic implant. However, ROS generated by local tissue cells and implant surfaces can contribute to implant failures [19]. These reactive oxygen intermediates produced at the implant–bone interfaces act as strong chemo-attractants, recruiting immune cells and leading to surrounding tissue damage and fibrosis (Figure 4). Additionally, ROS produced by immune cells can directly corrode implants. An imbalance between excessive ROS generation and insufficient antioxidant defense mechanisms reduces bone-implant osseointegration, further inducing aseptic implant loosening. This can lead to chronic inflammation, granulation tissue development, foreign body reaction, and fibrosis/fibrous capsule development, potentially leading to implant dysfunction.

Although implant debris can initiate an innate inflammatory response acting on numerous cells, including monocytes, fibroblasts, osteoblasts, osteoclasts, and mesenchymal stem cells (MSCs), resident macrophages play a crucial role in eliminating wear particles [20,21]. Macrophage activation is the dominant mechanism driving inflammation [22,23,24]. In some circumstances, the adaptive immune system may also be activated, particularly in response to metal ions associated with a hypersensitivity reaction to metals [25].

Therefore, there is a need for biomaterials with bioactive surface coatings that have antioxidant properties to improve implant osteointegration and stability, thus enhancing its effective lifespan. Several antioxidant strategies have been explored to reduce ROS formation in various prosthetic biomaterials. These strategies include surface functionalization, material doping with antioxidant agents, and nutritional supplementation [19].

Noninvasive methods are essential to examine and understand inflammation related to the implant [26]. Liposomes are particles of great interest for clinical applications. Due to their high biocompatibility and degradability, liposomes have been proposed as suitable nano/microparticles for delivering imaging or therapeutic agents.

The initial stages of acute inflammation are essential for bone healing. Liposomes have been used for optical imaging to study inflammation around implants in pre-clinical models. Riyaz et al. [1] employed sphingomyelin (SM) liposomes loaded with alpha-melanocyte-stimulating hormone (α-MSH) peptide in a rat model of bone implantation. α-MSH acts as a targeting ligand for macrophages and is an anti-inflammatory tridecapeptide cytokine derived from opio melanocortin that modulates macrophage reactivity. The researchers loaded the α-MSH-SM liposomes with an optical tracer, indocyanine green (ICG), to examine the effect of the debris released by the implant and its uptake by bone marrow-derived macrophages. The authors implanted a screw in the tibia of rats and used optical imaging post-implantation to assess the targeting efficacy and performance of the targeted liposomes. Comparisons were made between non-biodegradable (Ti) and biodegradable (Mg-10Gd) implants. As seen in Figure 5, the liposome’s ICG fluorescence was detectable at the inflammation site within one hour after administration, particularly in bone tissue, compared to muscle tissue during the early implantation phase.

The imaging results confirmed the suitability of the liposomes in the bone-implant system, showing clearance within 72 h. Notably, only the non-operated group showed significantly high signals in the bone at 24 h compared with the other operated groups (Figure 6). The comparison between non-biodegradable (Ti) and biodegradable (Mg-10Gd) implants highlights the potential benefits of using biodegradable materials. These materials may reduce long-term complications and the need for additional surgeries to remove implants.

Bone-resident tissue macrophages, also known as osteal macrophages, are myeloid cells that are different from osteoclasts. These osteal macrophages are located immediately next to osteoblasts and play diverse roles, such as regulating bone formation and maintaining skeletal homeostasis [27].

In the field of bone healing, liposomes have been studied for their potential to aid bone regeneration, both in the oral cavity and in broader orthopedic research. Researchers have utilized bone morphogenic protein 2 (BMP-2), a crucial biomolecule for bone regeneration. BMP-2, which is FDA-approved for human use, is recognized as the most potent molecule for osteoinduction [28]. Studies have demonstrated that incorporating BMP-2 into magnetic liposomes and delivering them via topical injection in a rat bone-defect model with an attached permanent magnet promotes new bone formation. This method of delivering BMP-2 magnetic liposomes, combined with magnetic implantation at the injury site, has shown promising results in enhancing bone regeneration [29].

## 6. Liposomes Detect Oxidative Stress and Enhance the Bone-Healing Process

Reactive oxygen species (ROS) are increasingly recognized as essential elements in the bone recovery process, influencing the physiological and pathological conditions of bone. But are they beneficial or harmful [30]? When ROS levels exceed physiological thresholds, they can cause cellular damage and contribute to disease development. Recent findings show that elevated ROS levels stimulate osteoclast differentiation and influence osteoblast formation, highlighting ROS’ significant role in regulating the human skeleton through redox signaling pathways. Innovative approaches such as using nanoplatforms to deliver therapeutic agents to bone tissue are being explored to treat osteoporosis [31,32].

Recent studies have highlighted a liposomal formulation as a contrast agent for hydrogen peroxide, a key marker for reactive oxygen species [33]. Researchers used so-called peroxyoxalate liposomes to detect hydrogen peroxide through chemiluminescence reaction. Hydrogen peroxide accumulates in cells, causing oxidative stress, which can be imaged with peroxyoxalate liposomes, potentially serving as both imaging and therapeutic agents.

Another example of ROS targeting and detection is the ROS Brite^TM^ 700 probe encapsulated into VCAM-1-targeted liposomes [34]. This probe produces bright near-IR fluorescence after ROS oxidation. The liposomes are used as nanocarriers to deliver specifically to the site of ROS formation within atherosclerotic plaques. This is a promising method for pre-clinical development of new therapeutic strategies or imaging approaches. It can also be used in other conditions marked by ROS activation and formation.

Additionally, researchers encapsulated 2,7-dichlorodihydrofluorescein (DCFH) within liposomes [35]. DCFH is non-fluorescent and becomes fluorescent upon oxidation by ROS, making it a valuable tool for detecting and quantifying ROS. Encapsulation of DCFH into liposomes offers protection against degradation and targeting capabilities.

Understanding redox signaling in bone health opens potential imaging and/or therapeutic targets for bone-related diseases.

## 7. Liposomes for Treatment of Fracture Healing

To address fracture healing issues, Zhou et al. [36] developed a salvianic acid A (CAS#: 7682-21-4)-loaded bone-targeting liposome formulation (SAA-BTL) using pyrophosphorylase cholesterol (cholesterol-PPi) as the targeting ligand. This formulation binds to the bone through the strong chelation of cholesterol-PPIs to bone apatite at various bone surfaces, including the growth plate, trabecular bone, and cortical bone. This results in a significant local distribution of high concentrations of SAA and enhanced retention, lasting longer than 20 days from a single injection in a glucocorticoid-induced delayed fracture model (HA). By regulating histone deacetylase 3 (HDAC_3_)-mediated endochondral ossification, SAA and SAA-BTL effectively promote osteogenesis and chondrogenesis during fracture healing, accelerating the conversion of cartilage to bone. HDAC_3_ is crucial for bone healing and development, being involved in endochondral ossification and cartilage development, which is a precursor to bone formation. Additionally, HDAC_3_ mediates inflammatory responses, which can influence the healing process.

## 8. Liposomes to Observe the Degradability of Implants

Helmholtz et al. [7] used fluorescently labeled sphingomyelin liposomes to study implantation-related cell stress. Fluorescence imaging was conducted on days 1, 3, 7, and 10 following the implantation of Mg or Mg-10Gd implants, with weekly scans after day 14. The liposomes were injected intravenously, and uCT was utilized to monitor implant degradation. The study revealed a significant material loss, with the Mg-10Gd alloy being the most degradable.

Over 6 weeks, the Mg and Mg-10Gd implants showed partial degradation, with mass changes of 14.4 (±1.7)% and 21.4 (±5.1)%, respectively. This corresponds to degradation rates of approximately 0.25 (±0.03) mm/a for Mg and 0.37 (±0.09) mm/a for Mg-10Gd, assuming homogeneous mass loss around the pin. These findings align with a previous study that also measured a slightly higher degradation rate for Mg-10Gd after bone implantation (Figure 7).

## 9. Discussion—Bioimaging

Approximately 8 to 10% of Americans have an implanted medical device, such as pacemakers, joint replacements, and bone implants. These devices are crucial for improving the quality of life for many individuals. However, they also present potential risks, including inflammation and oxidative stress at the implant site. These complications can lead to discomfort, impaired function, and even implant failure if not properly managed.

Liposomes, spherical vesicles composed of cholesterol and natural, non-toxic phospholipids, exhibit significant potential as imaging agents for detecting inflammation and oxidative stress complications. These vesicles can be synthesized to carry fluorescent markers or other imaging agents, making them highly effective for medical imaging. Liposomes can detect inflammation by functionalizing them to target specific cells involved in the inflammation process, such as macrophages. These cells are part of the body’s immune response and are often present in higher numbers at sites of inflammation. By targeting these cells, liposomes can provide a clear image of the inflamed area caused by bone implants.

Furthermore, liposomes can be engineered to respond to oxidative stress, a condition characterized by elevated levels of reactive oxygen species (ROS). ROS are chemically reactive molecules containing oxygen, increasing their levels in response to stress or injury. In the context of bone implants, oxidative stress can indicate potential complications. Liposomes designed to react with ROS could detect these areas of oxidative stress, offering valuable information about the health of the implant site.

Despite their potential, liposomes face several challenges, including rapid clearance from the body, precise targeting, immunogenicity, and production costs—issues which must be addressed. Ongoing research is focused on improving the design and functionality of liposomes to overcome these challenges. Enhancements in liposome technology aim to increase their stability, targeting accuracy, and overall effectiveness as diagnostic and therapeutic tools.

## 10. Conclusions

Using liposomes as imaging agents offers a noninvasive and highly sensitive method for monitoring bone implants. This approach enables early detection of complications and timely interventions, potentially improving patient outcomes and the longevity of the implants. By providing detailed imaging of inflammation and oxidative stress, liposomes can help healthcare providers make informed decisions about treatment and management.

Overall, using liposomes as imaging agents represents a promising strategy for advancing the monitoring and management of bone implants, ultimately enhancing patient care and implant success rates.

## 11. Search Strategy and Inclusion Criteria


Search Terms: bone implants, biodegradable implants, orthopedic implants, magnesium-based materials, bone remodeling, bone regeneration, bone-targeting, liposomes, functionalized-liposomes, bone-targeted drug delivery, bone-targeted imaging, inflammation in bone, macrophages and implants, macrophages and bone, reactive oxygen species in bone remodeling, liposomes and bone disease, non-invasive detection, inflammatory response, tissue response, imaging agents, liposomes and macrophages, liposomes and reactive oxygen species.Databases: PubMed, Google Scholar.Inclusion Criteria: Original and review papers were selected that provided information on the current issues with bone implants and possible interventions using nanotechnology, particularly liposomes. Keywords and search terms were used in searches. “And” and “or” were used for keyword searching. Full articles were saved and exported to the Mendeley Reference Manager Library. Articles from 2008 and until 2023 were included in this review.


## Figures and Tables

**Figure 1 cimb-47-00295-f001:**
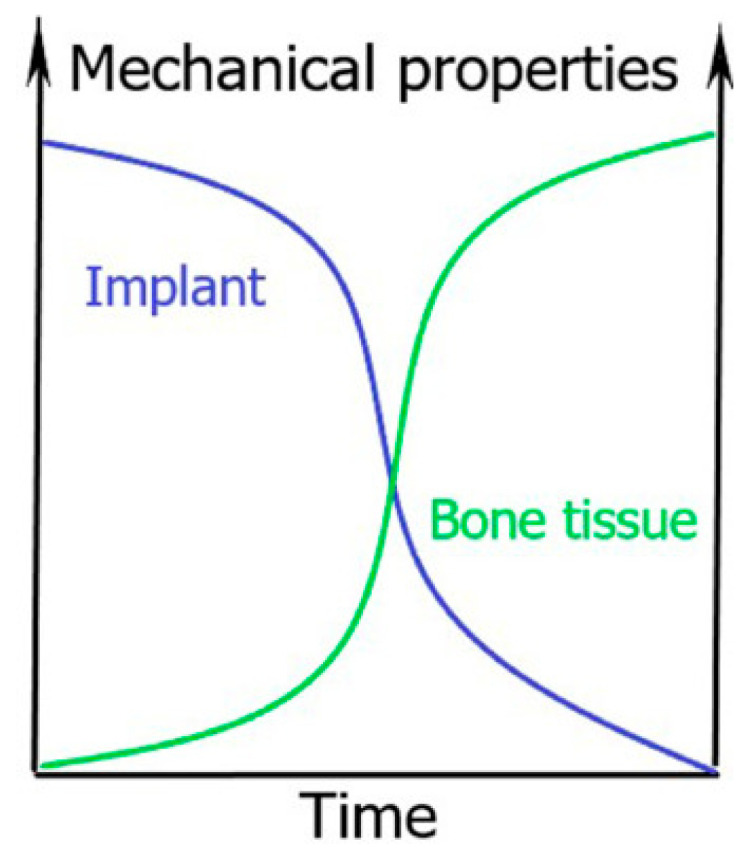
The ideal diagram represents the evolution of the implant over time (reduction of mechanical strength by degradation) simultaneously with the healing process of bone fracture [2].

**Figure 2 cimb-47-00295-f002:**
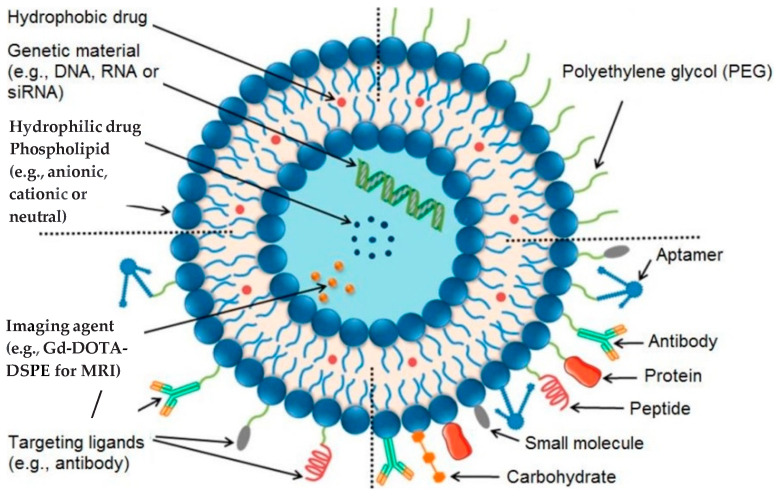
Structure of conventional and functionalized liposomes: (upper left) conventional liposomes comprising phospholipids; (upper right) PEGylated/stealth liposomes containing a layer of polyethylene glycol (PEG); (lower right) targeted liposomes containing a specific ligand to target a cancer site; and (lower left) multifunctional liposomes, which can be used for diagnosis and treatment of solid tumors. Adapted from [17] (Riaz et al., 2018).

**Figure 3 cimb-47-00295-f003:**
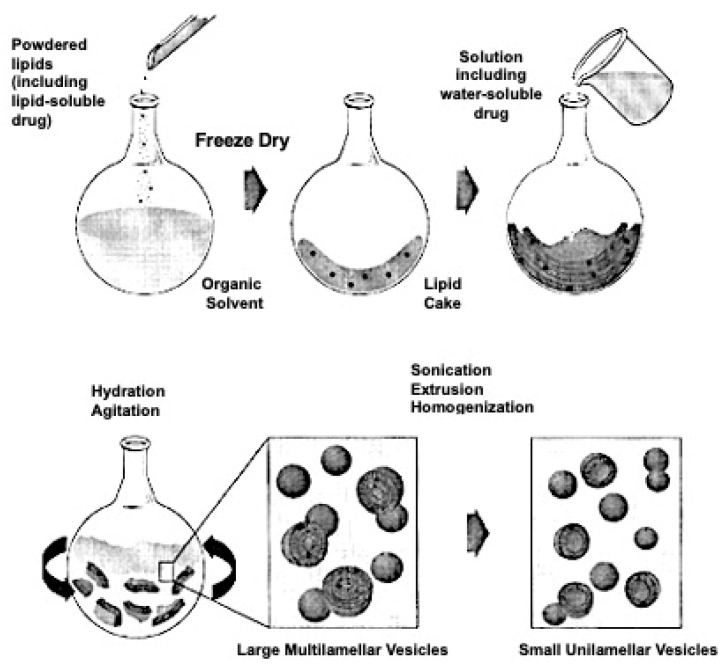
Liposome preparation (Avanti Research) [12].

**Figure 4 cimb-47-00295-f004:**
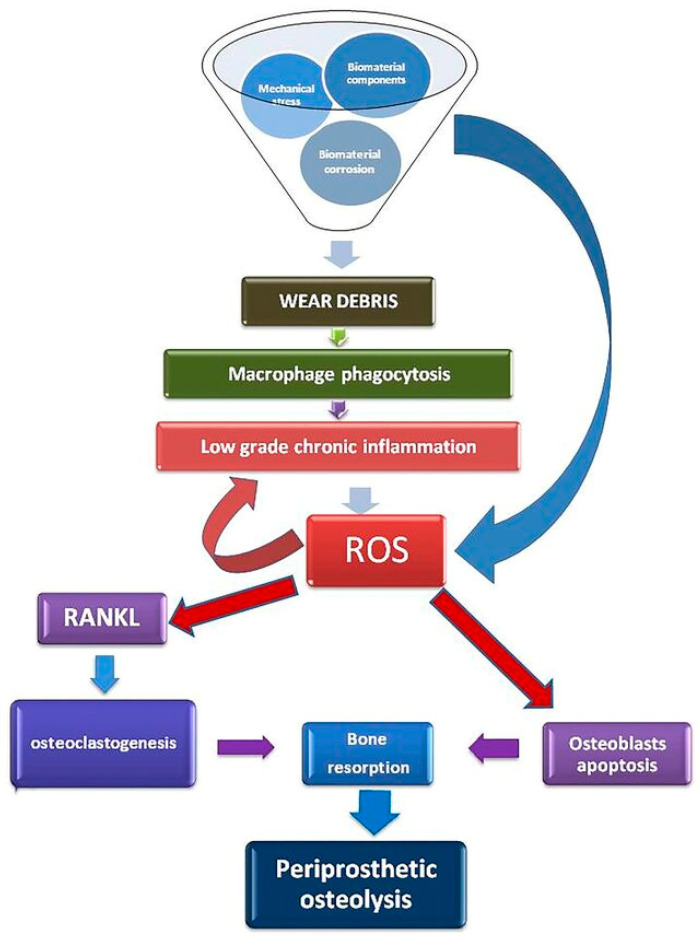
The inflammatory response to implants [19].

**Figure 5 cimb-47-00295-f005:**
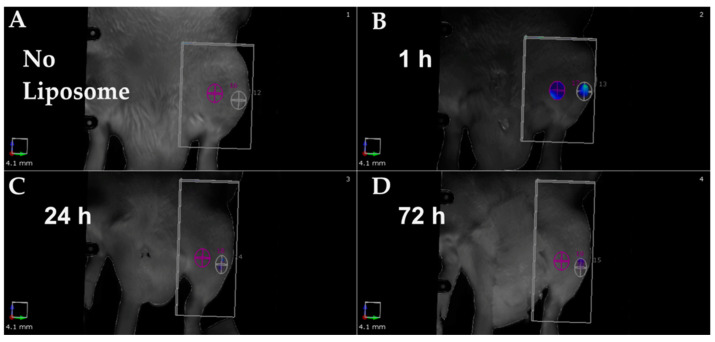
α-MSH-SM liposome usage for in vivo imaging of inflammation around the implant. The imaging was performed before liposome addition and at 1 h, 24 h, and 72 h after addition. The panel shows a representative image before fluorescent liposomes were injected (**A**) and post 1 h (**B**), post 24 h (**C**), and post 72 h (**D**) after liposome addition. The ROI was placed at the bone-implantation site with Mg-10Gd (white) and the muscle area (violet/colored) from which fluorescent signals were determined [1].

**Figure 6 cimb-47-00295-f006:**
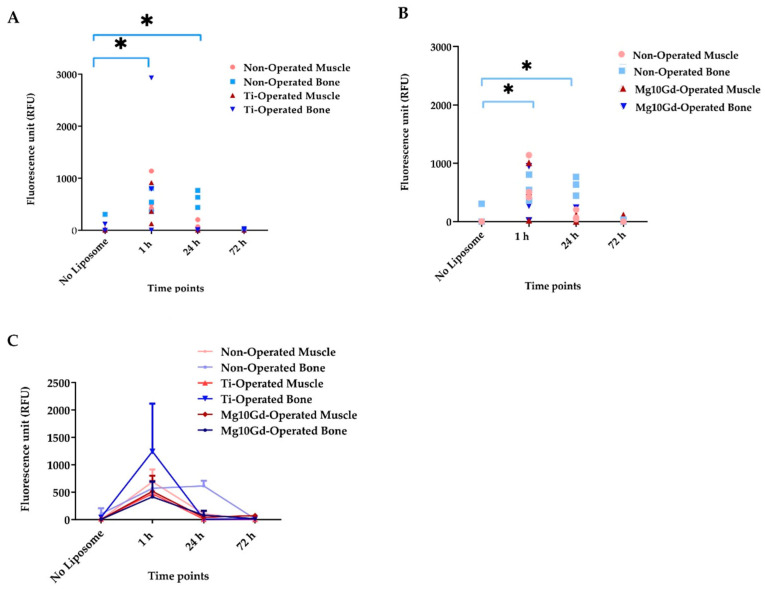
(**A**) Fluorescence intensities were compared between the non-operated group and the operated group with Ti screws. Data from each animal are presented. (**B**) Fluorescence intensity changes were compared between the non-operated and operated groups with Mg-10Gd screw. (**C**) Fluorescence intensities with the average signal from each group. Data are expressed as the mean ± standard error of the three samples per group: * *p* < 0.05 [1].

**Figure 7 cimb-47-00295-f007:**
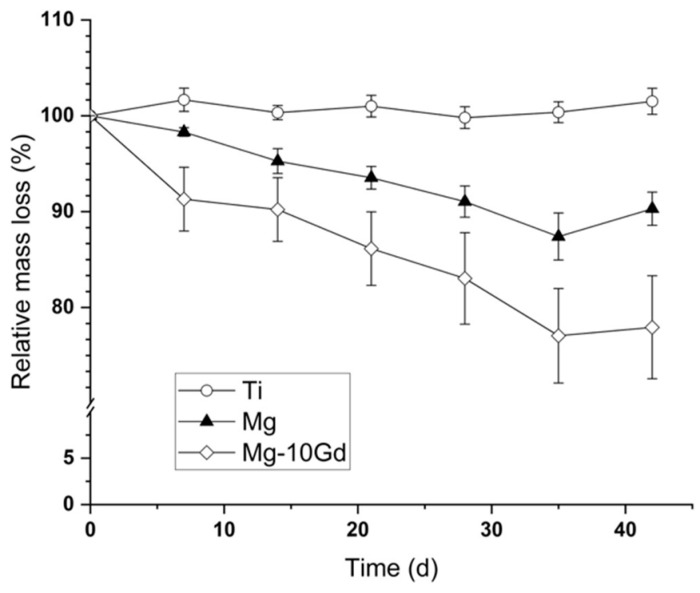
Relative mass loss of metallic implants over 42 days calculated from micro-computed tomography (μCT) observation; mean ± SEM, Ti n = 5; Mg and Mg-10Gd n = 10 [5].

**Table 1 cimb-47-00295-t001:** Characteristics of biodegradable Mg alloys used for orthopedic implants [2].

Advantages	Disadvantages
Good bioresorbability	High hydrogen release
Low density comparable with cortical bone (1.738 to 1.84 g/cm^3^)	In vitro degradation rate (407 mm per year)
Good strength-to-weight ratio	Fast degradation kinetics (implant degradation should occur concomitantly with bone remodeling)
Good biocompatibility	
High damping capacity	
Elastic modulus (41–45 GPa) is close to that of cortical bone	Low elastic modulus (implant must support loading without deformation)
Less stress shielding	High corrosion rate, thereby creating bio-incompatible environment in the surrounding tissues
Fracture toughness is greater than of ceramic biomaterials	Gas embolism
High machinability and dimensional accuracy

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
