# Peer review of "Liposomes as Imaging Agents of Inflammation and Oxidative Stress in Bone Implants"

_cimb, 2025, doi:10.3390/cimb47050295_

Round 1
Reviewer 1 Report
Comments and Suggestions for Authors
- The quality of all figures must improve
- In section 2. Imaging techniques that look at bone healing, the information is confusing. The title suggests that the authors will focus on imaging techniques but you described the implantation stages without the histology characteristics in health, injury, and inflammatory phases, you can include the tools for imaging analysis of bone treated with liposomes in vitro studies like STEM
- Section 5 Liposomes as imaging and therapeutic approaches in bone implants, this section must be improved you have to discuss transference factors (RUNX) and the sealing pathway involved with osteogenesis and osteoclastogenesis related to the NFkb inflammation pathway moreover you must include information about liposomes treatments and the change on specific proteins and mineralization bone. all this information can help to clarify the section 7. Liposomes detect oxidative stress and enhance the bone-healing process.
Author Response
Comments 1: In section 2. Imaging techniques that look at bone healing, the information is confusing. The title suggests that the authors will focus on imaging techniques but you described the implantation stages without the histology characteristics in health, injury, and inflammatory phases, you can include the tools for imaging analysis of bone treated with liposomes in vitro studies like STEM
Response 1: Thank you for pointing this out. We understand the concern. In order to discuss how imaging can be utilized for the evaluation of bone healing, the process involved after implantation would need to be discussed. We highlighted in green detailed information on the imaging techniques (page 2, lines 50 -51, and page 3, lines 60 – 74).
Comments 2: Section 5 Liposomes as imaging and therapeutic approaches in bone implants, this section must be improved you have to discuss transference factors (RUNX) and the sealing pathway involved with osteogenesis and osteoclastogenesis related to the NFkb inflammation pathway moreover you must include information about liposomes treatments and the change on specific proteins and mineralization bone. all this information can help to clarify the section 7. Liposomes detect oxidative stress and enhance the bone-healing process.
Response 2: We appreciate the reviewer’s thoughtful suggestion and the concern that was highlighted here is very well-taken. In an effort to improve clarity, we have combined sections 5 and 6 since they contained overlapping information. We hope that the approach to combining these sections will address your concerns.
Reviewer 2 Report
Comments and Suggestions for Authors
Dear Authors,
Thank you very much for providing the review article entitled “Liposomes as imaging agents of inflammation and oxidative stress in bone implants”.
The article reports an review of role of liposomes for imaging oxidative stress and inflammation in bone implants.
Please find below my comments regarding presented study:
Comments:
- The article lacks in description of methodology. No information on search, inclusion criteria or utilized databases is stated.
- Section 1 is entitled “General Introduction to Implants”. The emphasis is put specifically to biodegradable Magnesium alloy and it is not elaborated on other enlisted materials “…stainless steel, pure titanium, titanium-based alloys…”.
- Table 1 does not include references for a any of the presented information.
- Abbreviation MRI is not explained in the first place of appearance (Line 50). Please correct and check for every other abbreviation.
- Line 84: “The size of the nanoparticles should not exceed 200 nm” – Please justify.
- Line 122: “There are several ways to prepare liposomes” only one technique is described in the manuscript.
- Table 2 lacks context: clinically used liposome-based products are not described nor compared.
- Article lacks in in-depth conclusion as well as juxtaposing liposomes with other clinically applicable techniques of measuring oxidative stress and inflammation in bone implants.
General comment: the article is scientifically sound. Nevertheless it lacks structural organization and appears to present information selectively without in-depth description of various information (some of which are indicated in the comments above).
English language is acceptable. In the light of above comments I recommend the article for major revision.
Best regards,
Reviewer
Author Response
Comments 1: The article lacks in description of methodology. No information on search, inclusion criteria or utilized databases is stated.
Response 1: We appreciate this comment. The keywords used in the literature search were liposomes, non-invasive imaging agents, bone-implant complications, inflammation, oxidative stress, and bioimaging. These keywords were included in the submission on page 1, lines 20-21. The search was done in PubMed and Google/Google Scholar and includes articles from 2008 to 2023.
Comments 2: Section 1 is entitled “General Introduction to Implants”. The emphasis is put specifically to biodegradable Magnesium alloy and it is not elaborated on other enlisted materials “…stainless steel, pure titanium, titanium-based alloys…”.
Response 2: We agree with the reviewer’s comment. Even though we briefly introduced stainless steel, pure titanium, and titanium-based alloys, the focus was on biodegradable bone implants. As a result, we modified the title's content in Section 1 to reflect this (page 1, line 23).
Comments 3: Table 1 does not include references for any of the presented information.
Response 3: Thank you for this important note. We have provided a reference on line 45. As a result of this suggestion, the reference was also included on line 46 (page 2).
Comments 4: Abbreviation MRI is not explained in the first place of appearance (Line 50). Please correct and check for every other abbreviation.
Response 4: We appreciate this important inquiry. We have now included the full name before the abbreviation at the beginning of Section 2, line 50 (page 2).
Comments 5: Line 84: “The size of the nanoparticles should not exceed 200 nm” – Please justify.
Response 5: We understand your concern and we would like to point out that a rationale is now provided on lines 84 – 89 (page 3).
Comments 6: Line 122: “There are several ways to prepare liposomes” only one technique is described in the manuscript.
Response 6: We appreciate this important comment and have elaborated on the presentation of additional approaches to preparing liposomes. Such approaches are now listed on lines 126 through 131. Please note that line 122 has now become line 126 (page 4).
Comments 7: Table 2 lacks context: clinically used liposome-based products are not described nor compared.
Response 7: We would like to point out that the information in this table has now been discussed on lines 138 – 146 (page 5).
Comments 8: Article lacks in in-depth conclusion as well as juxtaposing liposomes with other clinically applicable techniques of measuring oxidative stress and inflammation in bone implants.
Response 8: We appreciate this comment. Therefore, we made some changes to the last section. The changes are highlighted from page 10 to page 11, specifically lines 307 – 308, lines 312 – 314, line 317, lines 324 to 329, and lines 331 to 335.
Round 2
Reviewer 2 Report
Comments and Suggestions for Authors
Dear Authors,
Thank you very much for providing the revised review article entitled “Liposomes as imaging agents of inflammation and oxidative stress in bone implants”.
Please find below my comments regarding presented study:
Comments:
- As per my last comment no.1 “The article lacks in description of methodology. No information on search, inclusion criteria or utilized databases is stated. “ the Authors replied “The keywords used in the literature search were liposomes, non-invasive imaging agents, bone-implant complications, inflammation, oxidative stress, and bioimaging. These keywords were included in the submission on page 1, lines 20-21. The search was done in PubMed and Google/Google Scholar and includes articles from 2008 to 2023”. The keywords in lines 20-21 identify your presented article and do not act as indication of search criteria for review article. Thank you for providing the information on search criteria. However, no information about it is still presented in the revised article. It is vital for review article to provide reproducible search strategy and inclusion criteria. Please rework article with respect to this comment.
- Please provide section regarding search strategy and inclusion criteria.
- Table 1: “Elastic modulus (41 – 45 GPa) is close to that of bone” – which type of bone are the Authors referring to? Please provide suitable reference and indicate type of bone.
- As per comment no.6 from last round of review: thank you for elaborating on different techniques. Nevertheless, the text fragment is lacking in references for different techniques and examples of liposomes obtained with each technique.
- No reference are provided for Figures 6, 7.
- In Section 9 please distinguish between discussion and conclusions.
Thank you for addressing the comments from previous review. I consider other, not mentioned, comments resolved. English language is acceptable. In the light of above comments I recommend the article for minor revision.
Best regards,
Reviewer
Author Response
We would like to thank the reviewers for their suggestions and comments. Please see below the responses to the comments.
Comments 1: As per my last comment no.1 “The article lacks in description of methodology. No information on search, inclusion criteria or utilized databases is stated. “ the Authors replied “The keywords used in the literature search were liposomes, non-invasive imaging agents, bone-implant complications, inflammation, oxidative stress, and bioimaging. These keywords were included in the submission on page 1, lines 20-21. The search was done in PubMed and Google/Google Scholar and includes articles from 2008 to 2023”. The keywords in lines 20-21 identify your presented article and do not act as indication of search criteria for review article. Thank you for providing the information on search criteria. However, no information about it is still presented in the revised article. It is vital for review article to provide reproducible search strategy and inclusion criteria. Please rework article with respect to this comment.
Comments 2: Please provide a section regarding search strategy and inclusion criteria
Response for both comments: Thank you for clarifying this. We corrected and answered this. This review paper now has a new section, section 11, titled Search Strategy and Inclusion Criteria. It covers the search terms, databases used, and inclusion criteria (page 11, lines 341 – 355).
Comments 3: “Elastic modulus (41 – 45 GPa) is close to that of bone” – which type of bone are the Authors referring to? Please provide suitable reference and indicate type of bone.
Response 3: Thank you for pointing out the omission. Reference 2 was provided for table 1. Tsakiris, V., Tardei, C., & Clicinschi, F. M. (2021). Biodegradable Mg alloys for orthopedic implants – A review. Journal of Magnesium and Alloys, 9(6), 1884–1905. https://doi.org/10.1016/j.jma.2021.06.024. Elastic modulus (41 – 45 GPa) is close to that of cortical bone. The word cortical was added to Table 1.
Comments 4: As per comment no.6 from last round of review: thank you for elaborating on different techniques. Nevertheless, the text fragment is lacking in references for different techniques and examples of liposomes obtained with each technique.
Response 4: We appreciate this comment. More clarity was added to this paragraph. The type of liposomes obtained by different techniques was included (page 4, lines 126 – 134).
Comments 5: No reference are provided for Figures 6, 7.
Response 5: We appreciate the comment. The manuscript contained a reference for Figure 6. It was reference 1 and can be seen at the end of the caption. We omitted the reference for Figure 7. We included it in this submission (line 302)
Comments 6: In Section 9 please distinguish between discussion and conclusions.
Response 6: We agree with the comment. As a result, we added Section 10—Conclusion to distinguish between discussion and conclusion.